# Evaluation of Aphicidal Effect of Essential Oils and Their Synergistic Effect against *Myzus persicae* (Sulzer) (Hemiptera: Aphididae)

**DOI:** 10.3390/molecules26103055

**Published:** 2021-05-20

**Authors:** Qasim Ahmed, Manjree Agarwal, Ruaa Al-Obaidi, Penghao Wang, Yonglin Ren

**Affiliations:** 1Agricultural Engineering Sciences, University of Baghdad, Al-Jadriya Campus, Baghdad 10071, Iraq; qasim.h@coagri.uobaghdad.edu.iq; 2College of Science, Health, Engineering and Education, Murdoch University, South Street, Murdoch, WA 6150, Australia; m.agarwal@murdoch.edu.au; 3Pharmacy College, Mustansiriyah University, Al-Qadisyia, Baghdad 10052, Iraq; ruaaloli@gmail.com

**Keywords:** aphids, *Myzus persicae* essential oils, synergistic activity, mortality of aphids

## Abstract

The insecticidal activities of essential oils obtained from black pepper, eucalyptus, rosemary, and tea tree and their binary combinations were investigated against the green peach aphid, *Myzus persicae* (Aphididae: Hemiptera), under laboratory and glasshouse conditions. All the tested essential oils significantly reduced and controlled the green peach aphid population and caused higher mortality. In this study, black pepper and tea tree pure essential oils were found to be an effective insecticide, with 80% mortality when used through contact application. However, for combinations of essential oils from black pepper + tea tree (BT) and rosemary + tea tree (RT) tested as contact treatment, the mortality was 98.33%. The essential oil combinations exhibited synergistic and additive interactions for insecticidal activities. The combination of black pepper + tea tree, eucalyptus + tea tree (ET), and tea tree + rosemary showed enhanced activity, with synergy rates of 3.24, 2.65, and 2.74, respectively. Essential oils formulation was effective on the mortality of aphids. Fourier Transform Infrared Spectroscopy (FTIR) analysis showed that stability of a mixture of essential oils was not affected by store temperature (15, 25, and 35 °C) and the functional groups were not changed during storage. Based on our results, the essential oils can be used as a commercial insecticide against *M. persicae*.

## 1. Introduction

Green peach aphid (GPA) or peach-potato aphid *Myzus persicae* (Sulzer) belongs to the family Aphididae in the order Hemiptera. This species is considered a polyphagous species that can be found worldwide and globally distributed. The GPA has been reported to feed on more than 500 species of plants from 40 plant families and is considered a major pest of many vegetable crops and plant families such as cucurbits, legumes, crucifers like cabbage, cauliflower, and broccoli, and solanaceous crops like potato, tomato, and capsicum. In addition, fruit crops such as peach trees, for example, serve as *M. persicae* hosts [1,2,3]. GPA damage comes from their feeding on plant sap, which causes yellowing and leaf curling of the plant. In addition, *M. persicae* has been involved in transmitting over 180 plant viruses. Hence, GPA, with a large host range exceeding 50 families of plants, is considered one of the most polyphagous aphids [1,4].

Plant-based biopesticides possessing insecticidal activities belonging to 60 families and showing promise as new botanical pesticides were reviewed and reported in several species of plants [5,6]. Most of the botanical pesticides have low to reasonable environmental toxicity, but there are exceptions, such as nicotine, and essential oils may degrade more rapidly in the environment than synthetic chemicals [7,8]. Many essential oils have caused high mortality of pests, were shown to be effective by use in different applications such as fumigation, and had antifeedant and repellent properties [9]. Moreover, some essential oils, such as cumin, anise, oregano and eucalyptus essential oils, showed aphicidal activity [10].

Black pepper *Piper nigrum* L. extracts showed insecticidal activities because they contain isobutyl amides that are toxic to insects. As reported in previous studies, piper extracts are a unique and valuable source of biopesticide [11,12], suitable for controlling small insects and reducing the development of pest resistance when mixed as a synergist with other botanical pesticides such as pyrethrum. It has been shown that black pepper essential oil reduces the adult emergence of cowpea weevil *Callosobruchus maculatus* by 100% after 30 days of treatment [13]. Furthermore, rosemary *Rosmarinus officinalis* essential oil has been used conventionally as a medication in many countries because it is non-toxic to humans and environments [14]. Moreover, it was reported that the effect of 1% rosemary oil on two-spotted mites *Tetranychus urticae* for contact toxicity on tomato caused high mortality after using the tomato leaf disc test for 12 and 48 h [14]. Rosemary essential oil was commercialized as a pesticide for its efficacy against several insect and mite pests; chemical compositions of rosemary oil have shown the LD_50_ values of the oil ranged from 58.9–335.9 μg/larva when applied to cabbage loopers *Trichoplusia ni,* and the LD_50_ values ranged 167.1–372.1 μg/larva for the fall armyworms *Pseudaletia unipuncta* [15,16,17]. Moreover, eucalyptus essential oil possesses a wide range of pesticide properties including insecticidal, insect repellent, herbicidal, acaricidal, fungicidal, and anti-microbial, and is considered non-polluting and environmentally friendly with little or no toxicological effect [2,18]. Eucalyptus essential oil has insecticidal properties in various forms such as contact, antifeeding, oviposition inhibition, repellence, and fumigant. Eucalyptus leaf extract is effective against aphids by the contact test method and antifeeding; it is effective against cotton leafhopper and cotton stainer by inhibiting oviposition. The fumigant of eucalyptus oil works against houseflies; and the oil and dry powder extract are used to protect potatoes against potato tuber moth [18,19]. In addition, tea tree *Melaleuca alternifolia* essential oil has been used as insect control agents because it contains bioactive chemicals that are toxic to several insect species, and researchers [20,21] have indicated that *M. alternifolia* may provide a new and safe alternative to chemical pesticides [15,20,21]. Many essential oils including tea tree oil have been examined on several hemipterans (aphids, thrips, whiteflies, and mealybugs) [22,23].

Essential oils represent a green alternative in agricultural fields due to reported insecticidal properties. Essential oils can be degraded by external factors such as temperature, light, and humidity. They have been analyzed by infrared spectroscopic techniques to discriminate function groups among different essential oils during storage [24].

This study was carried out to evaluate the insecticidal activities of essential oils against GPA *M. persicae* by studying contact toxicity and the synergistic activities of black pepper, tea tree, blue eucalyptus and rosemary essential oils because some pesticides are losing their effectiveness as a result of pest resistance. In addition, it evaluates a synergistic interaction between binary mixtures of essential oils against *M. persicae* in different methods of application. Fourier Transform Infrared Spectroscopy (FTIR) analysis was done to check the stability of essential oil mixtures during storage at different temperatures.

## 2. Results

### 2.1. Chemical Composition of Essential Oils

GC-MS analysis (Table 1) of black pepper (B), eucalyptus (E), rosemary (R), and tea tree (T) essential oils indicated that there are many major constituents in all the types of essential oils (pure and mixture): black pepper contained α-Pinene 12.66%, Sabinene 8.6%, β-Pinene 12.17%, 1R-α-Pinene 5.56%, D-Limonene 15.52%, Eucalyptol 3.21%, and Caryophyllene 24.56%, which were the most abundant compounds. The percentages of compounds in eucalyptus oil were P-Cymene 4.82%, D-Limonene 5.72%, and Eucalyptol 82.25%; whereas the main compounds in rosemary oil were α-Pinene 15.87%, Camphene 3.89%, β-Pinene 8.5%, P-Cymene 2.73%, D-Limonene 3.37%, Eucalyptol 35.27%, (−)-Camphor 10.43%, and Caryophyllene 4.92%. The proportions of tea tree chemical composition were (+)-4-Carene 7.31%, P-Cymene 3.8%, Eucalyptol 4.98%, γ-Terpinene 17.74%, Terpinolene 2.78%, (−)-Terpinen-4-ol 43.94%, and α-Terpineol 3.61%. However, the most abundant compounds in six types of essential oil combinations (black pepper + eucalyptus (BE), black pepper + rosemary (BR), black pepper + tea tree (BT), eucalyptus + rosemary (ER), eucalyptus + tea tree (ET), and tea tree + rosemary (TR)) were α-Pinene 6.76, 14.60, 7.95, 8.28, 1.59, and 9.28% found in BE, BR, BT, ER, ET, and TR, respectively. Camphene 0.23, 2.32, 0.25, 1.97, and 2.00% for BE, BR, BT, ER, and TR, respectively, with 1R- α-Pinene 2.62, 2.60, 2.85% found in BE, BR, and BT, respectively. Furthermore, (+)-4-Carene 3.51, 3.55, and 3.96% were found in BT, ET, and TR, while Sabinene 4.22, 4.27, and 4.58% were found in BE, BR, and BT, respectively. β-Pinene 6.12, 10.43, 6.75, 4.35, and 4.6% in BE, BR, BT, ER, and TR, respectively; whereas, P-Cymene 3.07, 1.94, 2.45, 3.81, 4.45, and 3.34%, and also D-Limonene 10.58, 9.34, 8.80, 4.56, 3.60, and 3.34% were found in all combination types. In addition, Eucalyptol 44.58, 19.63, 4.16, 59.45, 46.31, and 20.56% originated in six mixtures, and γ-Terpinene 8.50, 9.5, and 9.38% in BT, ET, and TR. The proportions of (−)-Terpinen-4-ol were 20.85, 20.94, and 22.02% in BT, ET, and TR, and the percentages of Caryophyllene in all combinations were 11.50, 14.62, 13.10, 2.36 and 2.73% for BE, BR, BT, ER, and TR, respectively.

### 2.2. Contact Toxicity of Pure Essential Oils

When essential oils were evaluated for their insecticidal activities using the contact bioassay method (Table 2), all black pepper, eucalyptus, rosemary and tea tree essential oils showed the mortality against *M. persicae*. The result indicated that black pepper oil exerted contact toxicity against GPA in a time and dose-dependent manner, and the dose 5% resulted in the highest mortality of aphids, followed by the dose 3% then 2% and 1%, compared with the untreated aphids at exposures of 1, 3, 6, 8 and 24 h. The highest mortality of 80% was observed for both black pepper and tea tree essential oil at 24 h exposure time; however, mortality at the concentrations of 1, 2 and 3 was low. Conversely, eucalyptus and rosemary were less effective on the target pest compared with black pepper and tea tree oils. The mortality with eucalyptus oil was 5, 8.33, 16.67, 33.33, and 53.33% at the concentration of 5% at 1, 3, 6, 8, and 24 h exposure times, and mortality from the rosemary oil at the dose 5% was 8.33, 8.33 (for 1 and 3 h), 23.33, 35, and 60%, respectively, at the same exposure time and R^2^ = 0.801 (Adjusted R^2^ = 0.795).

### 2.3. Contact Toxicity of Essential oil Mixtures

All the mixtures of essential oils tested were active toward aphids (Table 3). Some of the combinations of essential oils tested, such as BT, ET, and TR, caused more than 80% mortality in GPA at the maximum dose used after 24 h, while doses of 1 and 2% in some other combinations of essential oils showed less mortality than all types of combinations of essential oils. The contact activities of the three combinations TR, BT, and ET of essential oils (EOs) showed the highest mortality in the range of 93 to 98% at the concentrations of 3 and 5% for 24 h exposure time and R^2^ = 0.787 (Adjusted R^2^ = 0.78). However, the contact activities of other combinations of EOs were not ideal, causing less than 50–70% mortality during the 24 h exposure. The results of contact toxicity against GPA showed no significant differences between the EOs after 24 h exposure time.

### 2.4. Synergistic Activity among Essential Oil Mixtures

Insecticidal activity among four types of essential oils (black pepper, eucalyptus, rosemary and tea tree) were investigated (Table 4). The following EO combinations showed several synergistic interactions; the most significant synergy based on Wadley’s determination was the combination of BT, ET, and TR. the binary combination of two essential oils produced lower LC_50_ than the individual oil. The synergy interaction was found with the mixture of black pepper and tea tree, eucalyptus and tea tree, as well as tea tree and rosemary based on Wadley’s calculation, and the synergy ratio was *R* > 1.5. However, the combination of black pepper and eucalyptus, black pepper and rosemary, and eucalyptus and rosemary showed additive interaction because the synergy ratio was *R* < 1.5. No one of the essential oil mixtures showed antagonistic interaction *R* < 0.5 based on Wadley’s calculation.

### 2.5. Storage and Temperature Stability of Combined Essential Oils

The essential oil mixtures were stored at three different temperatures (15, 25, and 35 °C) for one, two, and three months, and all mixtures were introduced to FTIR analysis to determine functional group changes during the storage period. The functional groups present in the mixtures of essential oils were determined by comparing the vibration frequencies in the wave number. Figure 1, Figure 2, Figure 3, Figure 4, Figure 5 and Figure 6 present the spectral absorption of essential oil mixtures obtained from BE, BR, BT, ER, ET, and RT measured in the wave number range 4000–400 cm^-1^. The FTIR spectrum of BE showed strong broadband at 2922 cm^−1^ and was assigned to the Alkanes C-H stretch, a medium band intensity vibration at 1644 cm^−1^ was assigned to NH_2_, the strong methylene/methyl band at 1464.95 and 1446.16 cm^−1^. The peaks of IR at 2931 and 2842 cm^−1^ were due to Alkyl C-H asymmetric and symmetric stretching. The strong bands at 1631 cm^−1^ and 1633 cm^−1^ were indicative of aromatic compound C-H bending, at 1306–1361 cm^−1^ represented phenol compound –O-H and C-H stretch bands, while the number wave at 1015–1270 cm^−1^ showed the strong band of C-O. The position of the Alkenyl C=C stretching regularity varied slightly as a function of location around the double bond. Carbonyl compounds were often the strongest bands in the spectrum and were located between 1825 and 1575 cm^−1^. For a double bond, the function group played an important role in the observation of the carbonyl group. This included a connection of an aromatic group to a C=C or C=O. The wave number at 985 cm^−1^ indicated a strong or medium band for alkene compound C=C-H and at 920.1 cm^−1^ represented group C-H. For other oils no change in stretching and bending vibrations for C-H, C-O, C=C and O-H bonds were observed with respect to temperature and storage time.

## 3. Discussion

Plant essential oils are potentially valuable for GPA control. They performed in many ways on several types of insects and can be applied to crops or stored products [24]. Black pepper, eucalyptus, rosemary, and tea tree are known to possess antifeedant, repellent, ovicidal and insecticidal activities against many insect species [16]. Additionally, essential oils can be highly effective on insecticide-resistant insects as well as the use of chemical pesticides can be caused pesticides residues in the treated plants when used against insect pests [25,26]. In this study, the influences of essential oils varied according to oil type (pure oil and mixture), time, and dose on *M. persicae*. The insecticidal activity against GPA was observed with 1, 2, 3, and 5%. These essential oils might be important applicants for natural GPA control agents. All essential oils applied by contact proved toxic to *M. persicae*, although they differed in their efficacy. The two binary essential oil combinations resulted in higher mortality than the effect of a single essential oil. Synergistic combinations led to enhanced toxicity of the essential oil mixture and appeared to have multiple modes of action in pests compared with the use of a single essential oil, as shown in Tak et al. [26].

The GC-MS analysis (Table 1) showed that differences in chemical composition and its percentages were tested between essential oils in the pure and mixed oils. Black pepper GC-MS analysis showed that Caryophyllene was the most abundant major constituent followed by D-Limonene, α-Pinene, β-Pinene, and Sabinene. All these constituents affected various pests, and these results were consistent with [12,27]. The results indicated that Eucalyptol was one of the major chemical compositions in the eucalyptus essential oil, which made up (82.25%) of the total chemical composition of the oil. Eucalyptol has insecticidal activities against many insects, as shown in many previous studies [2,18]. In addition, the main constituents in rosemary essential oil were Eucalyptol, α-Pinene, (−)-Camphor, and β-Pinene. These components exhibit pesticide action that is used in pest control, as presented in many past experiments [6,7,14,27]. Moreover, the result showed that tea tree essential oil contained the highest compounds, which are (–)-Terpinen-4-ol and γ-Terpinene. These two major constituents have insecticidal properties and affect insect enzymes such as AChE, GST, and CarE [23,28,29]. However, the proportions of the main compounds in all types of tested essential oils changed after binary mixing. Chemical compositions in the BE combination were Caryophyllene, D-Limonene, α-Pinene, β-Pinene, and Sabinene. The result of GC-MS indicated variations of chemical compound percentages between the two pure essential oils as compared to their mixtures. In the BE mixture, the main constitutions were reduced to nearly half, compared with black pepper and eucalyptus separately (Table 1). The BE combination also showed most of the compounds found in the mixture that were not found in the pure oils, and this might have enhanced the range of insecticidal activity between oil constituents. Whereas in the BR combination, the highest compound amounts were α-Pinene, β-Pinene, D-Limonene, Eucalyptol, and Caryophyllene; these amounts were decreased compared with their purity before mixing. In BT, the result indicated that (–)-Terpinen-4-ol and Caryophyllene were the highest constituents in the mixture, this finding was consistent with [29], which indicated the high amounts of Terpinen-4-ol and Caryophyllene in tea tree *M. alternifolia* oil. In our study tea tree oil and binary mixtures that included tea tree oil (BT, ET, and TR) showed maximum efficacy against GPA, and this can be attributed to Terpinen-4-ol as the major constituent in all the above four oils. Moreover, ER, ET, and TR had the same components: α-Pinene (8.28, 1.59, and 9.28%, respectively), Eucalyptol (59.45, 46.31, and 20.56%, respectively), γ-Terpinene (1.36, 9.50, and 9.38, respectively) and (−)-Camphor (5.10, 0, and 5.32%). All these compounds have properties against various types of insects, as shown in previous studies [17,30].

The results of this study showed the insecticidal effects of black pepper, eucalyptus, rosemary, and tea tree essential oils and their binary combinations (BE, BR, BT, ER, ET, and TR) in various concentrations on the GPA. However, there were differences in the bio-insecticidal effects of 10 essential oils (four pure and six binary combination oils), despite them all having significant aphicidal activity on *M. persicae* in various mortality percentages based on the concentrations of essential oils, especially with high test doses for 24 h exposure time. The efficacy of essential oils depends upon their chemical composition and the proportion of each constituent present in the mixture. The essential oil constituents vary from one type to another, depending on plant species [7]. Hollingsworth et al. [31] and Liška et al. [32] reported that α-Pinene, D-Limonene, and Camphene, which were major constituents, demonstrated aphicidal properties against the wooly beech aphid *Phyllaphis fagi* and the palm aphid *Cerataphis brasiliensis*. Conversely, previous studies showed that limonene is able to attract aphids’ natural enemies, so essential oil containing a high amount of D-Limonene can act as an attractant for parasitoids and predators [30], while the compounds Eugenol and 1,8-Cineole have been reported effective against many insects, including aphids [31,32,33]. Other major constituents, such as Terpinene-4-ol, Terpinene, 4-Carene, and α-Phellandrene also have an insecticidal effect against several insects, and these compounds were found in the tested essential oils (black pepper, eucalyptus, rosemary, and tea tree) [34,35]. Some previous work related to the aphicidal effects of rosemary oil were consistent with our findings. Rosemary oil has insecticidal effects against several insects and is used in many commercial products as an insecticide [25]. A study conducted by Digilio et al. [36] found that rosemary is a repellent and contact toxin against *M. persicae* and has a greater ability to penetrate through the cuticle of aphids than to be absorbed from the gut and intestines. These findings are similar to the results presented by Tomova et al. [37], indicating an increase in the mortality percentage of aphids when exposed to diverse essential oils; Görür et al. [38] confirmed the significant effects of the essential oil volatiles against three species of aphids, demonstrating a potential for aphid control. Black pepper and tea tree essential oil contact application resulted in 80% mortality in the GPA; there is a report of comparable findings with results of nearly 100% mortality caused by black pepper and tea tree essential oils against the rose-grain aphid [22]. In addition, our findings indicated that the mortality rates caused by eucalyptus and rosemary oils were 94.44 and 95.56%, respectively (Table 2 and Table 3). However, the combination of TR, ET, and BT essential oils can cause 98.33, 95.00, and 98.33% mortality, respectively, while contact application showed that the parallel effect of essential oils against cabbage aphids resulted in more than 85% mortality [39]. Conversely, it was concluded that there was an impact on the aphid population on the plant when exposed to a low dose [39]. Our findings on the insecticidal effects of the tested combined essential oils showed high mortality rates caused by TR, ET, and BT, and more than 96% mortality for BR, BT, and ET. In the series of concentrations applied, mortality was concentration-dependent. The reason for these results was the essential oils’ constituents and their volatiles. Our result was consistent with [37], who showed high mortality rates in *M. persicae* and *Acyrthosiphon pisum* by using different doses of various essential oils (Table 2 and Table 3). Regarding the use of these oils, it can be concluded that black pepper and tea tree have stronger insecticidal potential than eucalyptus and rosemary.

We observed that individual oils varied in their toxicity to the aphids and some oil types were more toxic to aphids, especially in a high dose. The mixed binary essential oils were also shown to be additive, synergistic, and no combinations were antagonistic. We found their toxicity was as high as expected, especially with a short time and a low dose. The toxicity of the binary mixtures of oils increased the mortality of aphids. This indicated that some combinations had a synergistic effect; however, synergistic activities were observed in the BT, ET, and TR combinations, LC_50_ values were 1.57, 2.36, and 2.23, respectively, compared with individual LC_50_ values of 5.16, 8.27, 5.03, and 7.76 for B, E, T, and R, respectively (Table 4). In most mixtures, additive interaction was observed between two mixed oils such as BE, BR, and ER, and that happened because the binary combination may have returned to the interaction between the essential oils’ components [40]. Synergistic interaction in the cases of BT, ET, and TR against *M. persicae* can be attributed to: (1) the difference in the insecticidal mechanisms of black pepper and tea tree from other essential oils on *M. persicae*; and (2) the synergistic effect being due to the essential oils’ compounds interaction. These findings were parallel to Pavela [41], who reported that synergistic insecticidal activities could be observed not only in the combination of essential oils but also in their constituents, as well as between synthetic insecticides with essential oils.

The findings of our study indicated that the three tested temperatures had no effect on essential oil stability based on FTIR analysis, as no changes in functional groups could be seen in all periods of storage. Our results were consistent with Turek and Stintzing [24], who demonstrated oil stability at various temperatures (5–38 °C), and that it showed no effect from temperature for three months when stored. All these peaks of function groups were not affected by the storage temperature tested on the essential oils or the time of storage, and all groups remained at the same wavenumber location. FTIR analysis was performed because is a fast and relatively cheap technique that might allow direct function group measurements of components in mixtures.

## 4. Materials and Methods

### 4.1. Essential Oils and Chemicals

Black pepper *Piper nigrum*, rosemary *Rosmarinus officinalis*, eucalyptus blue gum *Eucalyptus globulus,* and tea tree *Melaleuca alternifolia* 100% pure essential oils were obtained from Essential Pure Natural Select Ingredient supplies (Range Products Pty. Ltd., Perth, Western Australia), which were extracted by steam distillation (Table 5). Methanol 99.9% and hexane 97% were purchased from Sigma-Aldrich (Australia), and acetone 99.5% and ethanol 99.0% were purchased from Asia Pacific Specialty Chemicals Ltd., NSW, Australia.

### 4.2. Aphid Rearing

GPA *Myzus persicae* (Sulzer) were obtained from two locations, Agricultural Biotechnology Centre (SABC) at Murdoch University and the field (GPS Coordinates: 32.0699° S, 115.8426° E), Western Australia. Green peach aphids were reared in a glasshouse located at Murdoch University, Western Australia, on cabbage *Brassica oleracea* var. *capitata* and sweet capsicum *Capsicum annuum* potted plants under the glasshouse condition for aphid colonies: the temperature ranged between 18 ± 2 °C and 25 ± 2 °C during daylight and at night with humidity between 60 ± 2% and 75 ± 2%, respectively, and a photoperiod L18:D6. The temperature and humidity were recorded using the HoBoware^®^ (temperature/Relative Humidity data logger) and its data loggers software version 3.7.18 (Onset Company, One Temp Pty. Ltd., Adelaide, Australia), held inside the glasshouse. *M. persicae* were transferred by a fine brush and placed on the leaves of cabbage and capsicum plants.

### 4.3. Determination of Essential Oil Compounds Using Gas Chromatography-Mass Spectrophotometry (GC-MS)

GC-MS analysis was carried out for 100% pure essential oil of black pepper, eucalyptus blue gum, rosemary, and tea tree, and their binary combinations by using a Shimadzu GC-MS model QP2010 series, installed with an SGE main category BPX5 column, using 30 m × 0.25 mm film thickness 0.25 μm (Kinesis Australia Pty Ltd., Qld, Australia) and AOC-5000 autosampler (Shimadzu, Kyoto, Japan) as an autosampler. The parameter of GC-MS analysis used the following method: injector temperature 220 °C; pressure 63.43 kPa; column flow 1.07 mL/min; linear velocity 37.8 cm/s; sample injected volume 1 μL diluted with hexane. Gas chromatography coupled to a mass selective detector (MSD) were recorded with ionization and interface temperature of 200 °C; the solvent cut time was 1.5 min and the carrier gas helium. Two replicates for each essential oil were injected. The individual constituent of each essential oil was identified and achieved by comparing the obtained mass spectra for each component with the values stored in mass spectra libraries and the NIST database with data previously reported in the literature. The percentage composition of the oils was calculated in peak areas using the normalization method.

### 4.4. Contact Toxicity Bioassay of Pure and Combination Essential Oils

Four concentrations were used to test the contact toxicity for 10 essential oils, which were four pure essential oils of black pepper, eucalyptus blue gum, rosemary, and tea tree and six binary mixtures of essential oils of black pepper + eucalyptus, black pepper + rosemary, black pepper + tea tree, eucalyptus + rosemary, eucalyptus + tea tree, and rosemary + tea tree (BE, BR, BT, ER, ET, and TR). Three replications were used per treatment of essential oil for the pure and mixture. The control treatment was treated as above but using the solvent only. Between 20 and 30 aphids (different stages of aphids) were placed on filter paper in a 9 cm Petri dish. Each treatment was sprayed by using a micro-spray size 5 mL sprayer, and 1 mL of essential oil was applied according to each concentration. Petri dishes were covered with mesh and placed in an incubator chamber (25 ± 2 °C, 16:8, L:D, 65 ± 5% RH). Mortality was determined under a microscope after 1, 3, 6, 8, and 24 h. Three replicates were used for each concentration.

### 4.5. Synergistic Interactions between Essential Oils

To evaluate potential synergies between the essential oils (black pepper, eucalyptus blue gum, rosemary, and tea tree), mixtures were prepared maintaining the same concentration of single essential oil, following 1:1 ratio of the oils. Mixtures were applied to aphid adults and their LC_50_ values were estimated after 24 h. The relationships of the mixtures were determined by using two statistical models, which were Hewlett and Plackett’s model and Wadley’s model, to compare expected and observed LC_50_ values as shown in Equations as per Tak et al. [26]. Depending on Hewlett and Plackett’s calculation, the expected LC_50_ values (assuming additive interaction) were determined from:(1)$$E=\left(a\times {\mathrm{LC}}_{50}\left(a\right)\right)+\left(b\times {\mathrm{LC}}_{50}\left(b\right)\right)+\left(c\times {\mathrm{LC}}_{50}\left(c\right)\right)+\cdot \cdot +\left(n\times {\mathrm{LC}}_{50}\left(n\right)\right)$$
where E refers to expected LC_50_ and a is the proportion of oil A in the mixture. LC_50_ (a) is the LC_50_ of oil A and b is the proportion of oil B in the mixture, as well as LC_50_ (b) is the LC_50_ of oil B and the rest according to Wadley, theoretical LC_50_ values were calculated from:(2)$$E=\frac{a+b+c+\ldots +n}{\frac{a}{{\mathrm{LC}}_{50}\left(a\right)}+\frac{b}{{\mathrm{LC}}_{50}\left(b\right)}+\frac{c}{{\mathrm{LC}}_{50}\left(c\right)}+\ldots +\frac{n}{{\mathrm{LC}}_{50}\left(n\right)}}$$
where E, a, b, c …. and n are as described above. The interaction between the observed and theoretical LC_50_ values (Equation (3)) was compared:(3)$$R=\frac{{\mathrm{Expected}\;\mathrm{LC}}_{50}}{{\mathrm{Observed}\;\mathrm{LC}}_{50}}$$
where *R* represents synergistic interaction; the relationship between the constituents of the mixture is defined as either synergistic (when *R* > 1.5), additive (1.5 ≥ *R* > 0.5), or antagonistic (*R* ≤ 0.5), based on this model.

### 4.6. Stability of Essential Oils and Their Combinations at Different Time Intervals by Fourier Transform Infrared Spectroscopy (FTIR)

The FTIR spectrum of the mixture of essential oils was performed on a Perkin Elmer Spectrum Version 10.4.2 model Frontier FTIR/NIR in the School of Engineering and Information Technology, Murdoch University. Functional groups were determined with the help of IR correlation charts. IR spectra were shown in the percentage of absorbance and the wave number region for FTIR analysis from 4000–400 cm^−1^. The FTIR software (version 2.3.1.5) and the OMNIC window with ATR cell were used for the analysis of the states of chemical bonding. The number of scans and their resolution were four with a resolution of 4 cm^−1^, and the detector MIR TGS (Waltham, Massachusetts, USA) was used. Beamsplitter OptKBr, Apodization strong, spectrum type spectrum, beam type ratio, phase correction magnitude, scan speed 0.2, IGram type Double, scan direction Combine JStop 8.94, IR-Laser Wavenumber 15798.00, Description DATR 1 bounce Diamond/KRS5.

The combinations of essential oil samples were prepared by mixing different binary types of essential oils in the ratio (1:1) and storing at three different temperatures, which were 15, 25, and 35 °C, for one, two, and three months. The FTIR was conducted after three months to determine the essential oil contents’ stability of functional groups for the combination. A small drop of around 1 mL of the essential oil was placed on the plate and the spectrum run. The plates were thoroughly cleaned after each scan of an essential oil to prevent contamination of other samples and the diamond wiped with a tissue, then washed several times with methylene chloride, then ethanol, to remove the sample. The cleaned surface was clear and free from scratches.

### 4.7. Data Analysis

Mortality data from the essential oil elimination assay were subjected to analysis of variance (ANOVA) by using SPSS software version 24.0 (IBM Crop, Armonk, NY). Aphids were considered to be dead when no movement was detected by checking with a needle under a magnifying glass. Probit analysis was used to calculate the lethal dose (LC_50_) values that caused 50% mortality compared with the untreated aphids by using MS Chart software version 2016.12.07 [42]. Microsoft Office Excel version 2016 was used to analyze FTIR and GC-MS data.

## 5. Conclusions

In conclusion, the current study demonstrates the strong toxic effects against aphids using the essential oils of black pepper, eucalyptus, rosemary, and tea tree, and their binary combinations. They were found to have effective insecticidal properties in contact toxicity on GPA. The contact treatment of pure essential oil indicated that black pepper and tea tree essential oils were more effective than eucalyptus and rosemary essential oils on aphids at a high concentration after 24 h. Essential oils are natural plant products containing a complex mixture of compounds and thus have multiple insecticidal or aphicidal properties. The essential oil mixtures showed their insecticidal effects on aphids and the interaction between the binary oils led to synergistic, additive effects. There was a synergistic effect between black pepper and tea tree, eucalyptus and tea tree, and rosemary and tea tree essential oils, while the other combinations of black pepper and eucalyptus, black pepper and rosemary, and eucalyptus and rosemary showed additive interactions. According to the FTIR analysis, essential oil combinations were stable between 15–35 °C without affecting the properties of the oils. Therefore, we suggest that tested essential oil constituents in a pure state and in combinations should be screened as potential natural insecticides or be included in the chemical synthesis of a new type of pesticide, based on essential oils and their constituents.

## Figures and Tables

**Figure 1 molecules-26-03055-f001:**
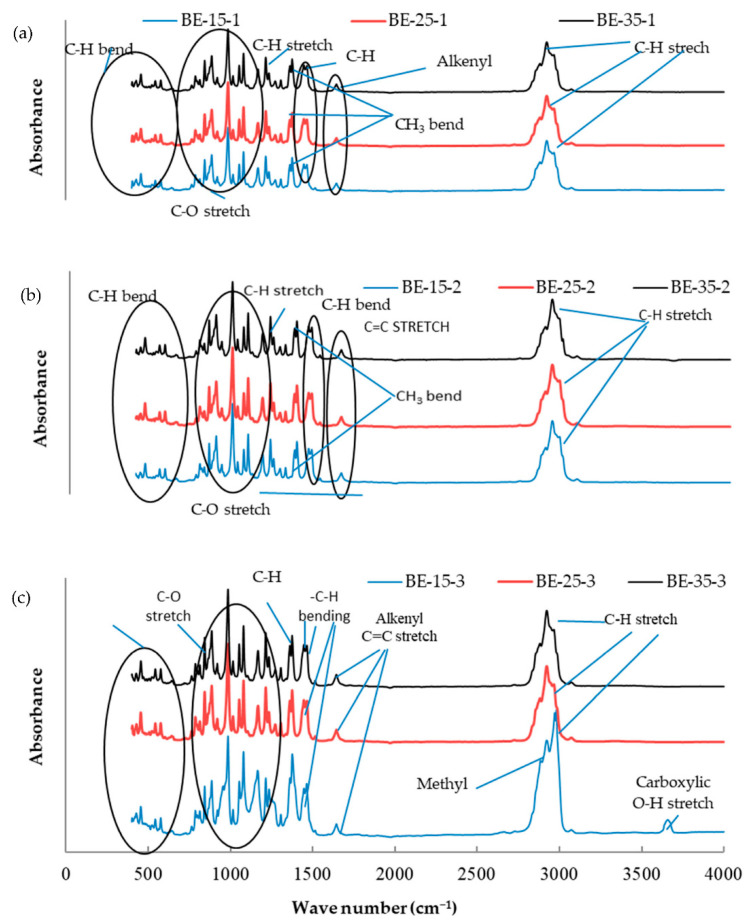
Fourier-transform infrared spectroscopy (FTIR) analysis for the essential oil combination of black pepper + eucalyptus (BE) at different temperatures (15, 25, and 35 °C) and different times of storage (1–3 months) (different colors refer to different temperature per subfigure). (**a**) This figure refer to the effect of different temperature at 15, 25, and 35 °C on the combination of BE stored for one month; (**b**) the second subfigure refer to different temperature at 15, 25, and 35 °C on the combination of BE stored for two months; (**c**) the third subfigure refer to different temperature at 15, 25, and 35 °C on the combination of BE stored for three months.

**Figure 2 molecules-26-03055-f002:**
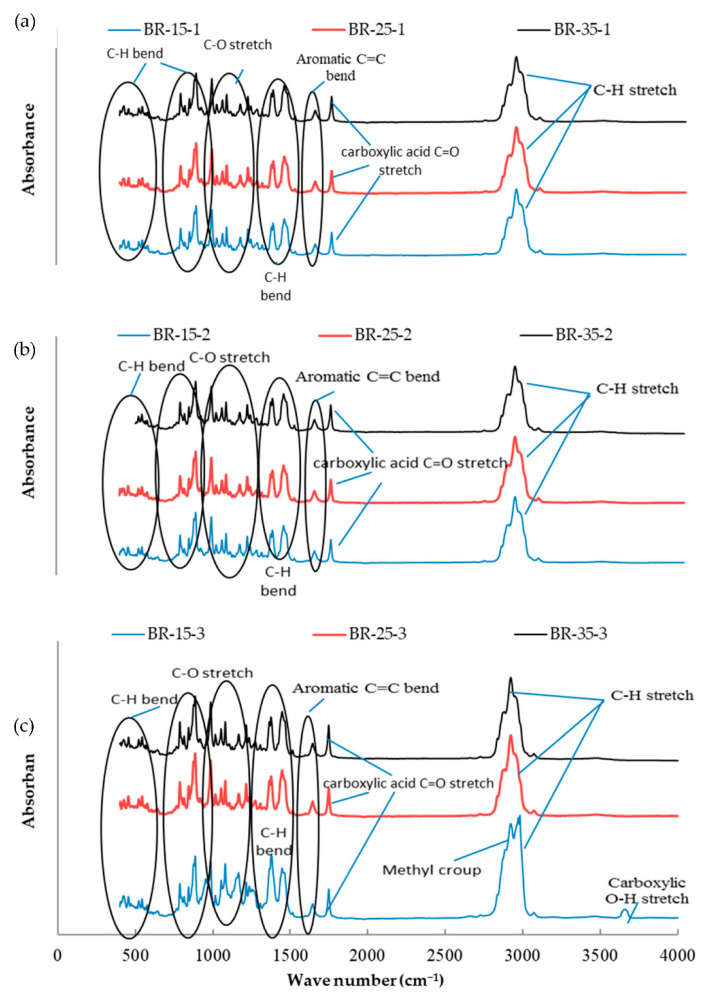
Fourier-transform infrared spectroscopy (FTIR) analysis for the essential oil combination of black pepper + rosemary (BR) at different temperatures (15, 25, and 35 °C) and different times of storage (1–3 months) (different colors refer to different temperature per subfigure). (**a**) This figure refer to the effect of different temperature at 15, 25, and 35 °C on the combination of BR stored for one month; (**b**) the second subfigure refer to different temperature at 15, 25, and 35 °C on the combination of BR stored for two months; (**c**) the third subfigure refer to different temperature at 15, 25, and 35 °C on the combination of BR stored for three months.

**Figure 3 molecules-26-03055-f003:**
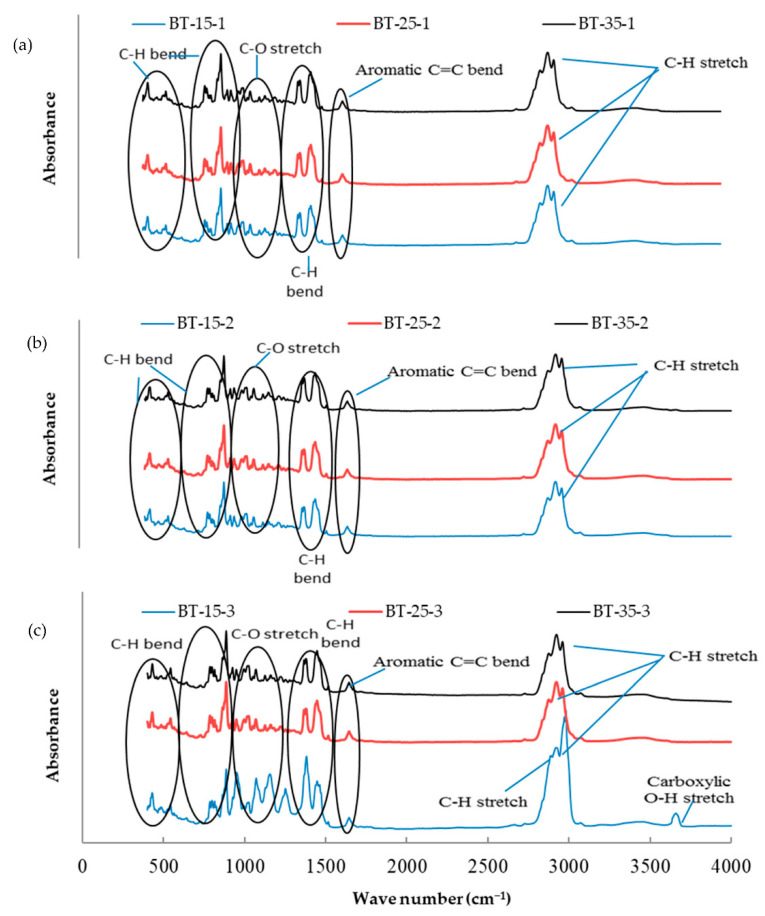
Fourier-transform infrared spectroscopy (FTIR) analysis for the essential oil combination of black pepper + tea tree (BT) at different temperatures (15, 25, and 35 °C) and different times of storage (1–3 months) (different colors refer to different temperature per subfigure). (**a**) This figure refer to the effect of different temperature at 15, 25, and 35 °C on the combination of BT stored for one month; (**b**) the second subfigure refer to different temperature at 15, 25, and 35 °C on the combination of BT stored for two months; (**c**) the third subfigure refer to different temperature at 15, 25, and 35 °C on the combination of BT stored for three months.

**Figure 4 molecules-26-03055-f004:**
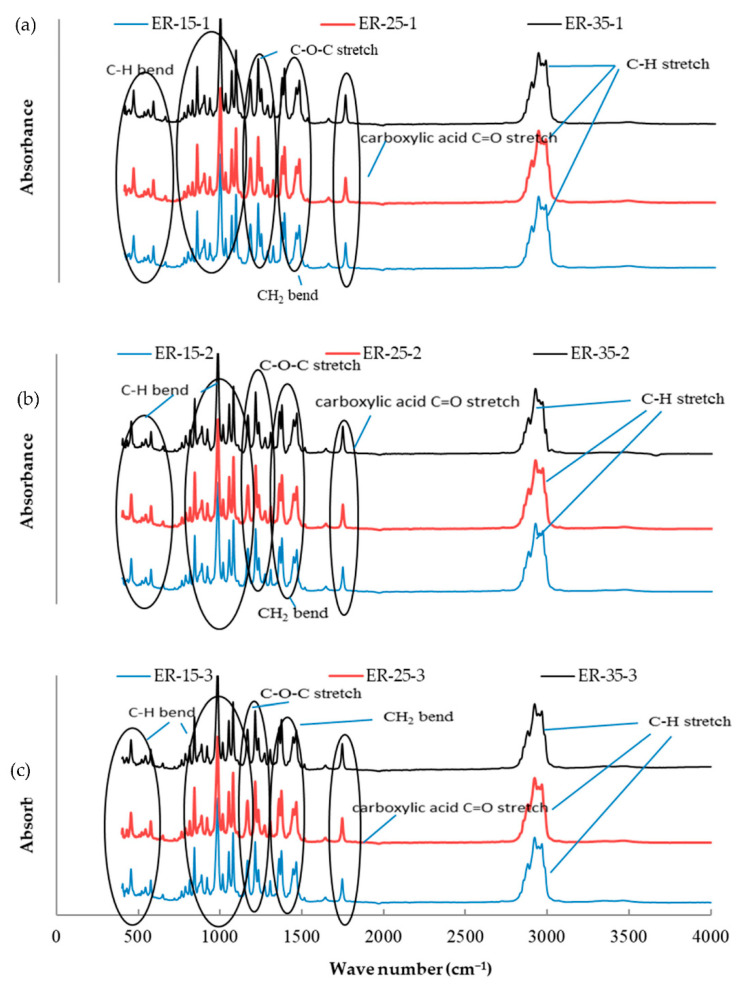
Fourier-transform infrared spectroscopy (FTIR) analysis for the essential oil combination of eucalyptus + rosemary (ER) at different temperatures (15, 25, and 35 °C) and different times of storage (1–3 months) (different colors refer to different temperature per subfigure). (**a**) This figure refer to the effect of different temperature at 15, 25, and 35 °C on the combination of ER stored for one month; (**b**) the second subfigure refer to different temperature at 15, 25, and 35 °C on the combination of ER stored for two months; (**c**) the third subfigure refer to different temperature at 15, 25, and 35 °C on the combination of ER stored for three months.

**Figure 5 molecules-26-03055-f005:**
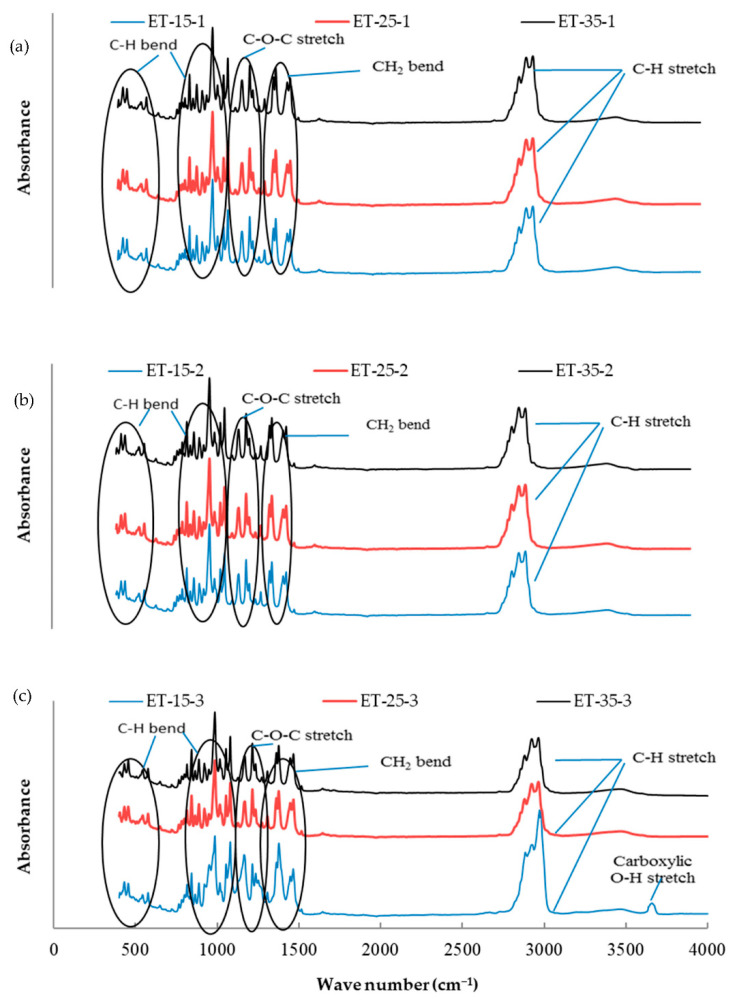
Fourier-transform infrared spectroscopy (FTIR) analysis for the essential oil combination of eucalyptus + tea tree (ET) at different temperatures (15, 25, and 35 °C) and different times of storage (1–3 months) (different colors refer to different temperature per subfigure). (**a**) This figure refer to the effect of different temperature at 15, 25, and 35 °C on the combination of ET stored for one month; (**b**) the second subfigure refer to different temperature at 15, 25, and 35 °C on the combination of ET stored for two months; (**c**) the third subfigure refer to different temperature at 15, 25, and 35 °C on the combination of ET stored for three months.

**Figure 6 molecules-26-03055-f006:**
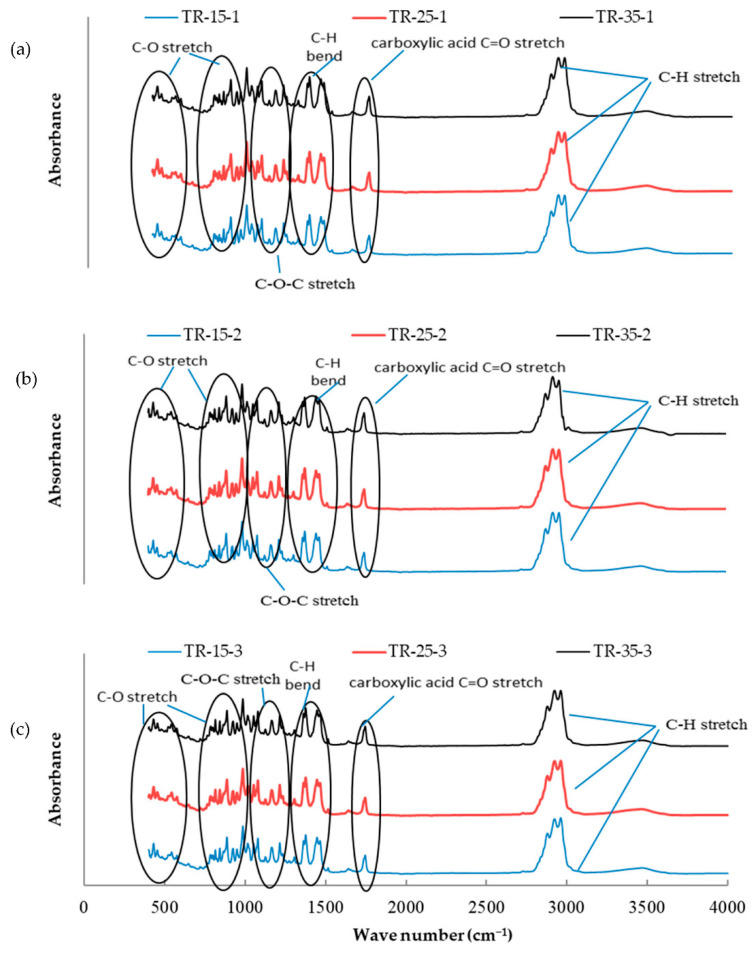
Fourier-transform infrared spectroscopy (FTIR) analysis for the essential oil combination of tea tree + rosemary (TR) at different temperatures (15, 25, and 35 °C) and different times of storage (1–3 months) (different colors refer to different temperature per subfigure). (**a**) This figure refer to the effect of different temperature at 15, 25, and 35 °C on the combination of TR stored for one month; (**b**) the second subfigure refer to different temperature at 15, 25, and 35 °C on the combination of TR stored for two months; (**c**) the third subfigure refer to different temperature at 15, 25, and 35 °C on the combination of TR stored for three months.

**Table 1 molecules-26-03055-t001:** Chemical constituents of pure and mixed essential oils analysis by GC-MS.

RT ^a^ (min)	Composition	Percentage (%) of Essential Oil Composition ^b^									
		B	E	R	T	BE	BR	BT	ER	ET	TR
1.97	Cyclohexane	2.89	2.72	2.56	2.80	2.92	2.57	2.83	2.59	2.48	2.87
5.73	Tricyclene	-	-	0.66	-	-	0.32	-	0.32	-	0.34
5.77	α-Thujene	0.89	-	-	0.76	0.49	0.51	0.91	-	0.36	0.36
6.07	α-Pinene	12.66	0.90	15.87	2.27	6.76	14.60	7.95	8.28	1.59	9.28
6.54	α-Fenchene	-	-	0.10	-	-	-	-	-	-	-
6.60	Camphene	0.47	-	3.89	-	0.23	2.32	0.25	1.97	-	2.00
7.38	Sabinene	8.60	-	0.18	-	4.22	4.27	4.58	-	-	0.10
7.60	β-Pinene	12.17	0.35	8.50	0.58	6.12	10.43	6.75	4.35	0.47	4.60
7.94	β-Myrcene	1.00	0.44	1.47	0.55	0.73	1.26	0.81	0.95	0.50	1.02
8.72	α-Phellandrene	0.20	0.41	0.26	0.28	0.41	0.35	0.34	0.34	0.37	0.29
8.79	1R-α-Pinene	5.56	-	-	-	2.62	2.60	2.85	-	-	-
9.16	(+)-4-Carene	-	-	0.58	7.31	-	0.30	3.51	0.30	3.55	3.96
9.56	p-Cymene	1.10	4.82	2.73	3.80	3.07	1.94	2.45	3.81	4.45	3.34
9.73	d-Limonene	15.52	5.72	3.37	1.10	10.58	9.34	8.80	4.56	3.60	2.30
9.88	Eucalyptol	3.21	82.25	35.27	4.98	44.58	19.63	4.16	59.45	46.31	20.56
10.47	Z-Ocimene	0.05	-	-	-	-	-	-	-	-	-
11.02	γ-Terpinene	-	1.71	0.99	17.74	0.92	0.54	8.50	1.36	9.50	9.38
12.47	Terpinolene	0.11	-	-	2.78	-	-	1.39	-	1.37	1.39
13.28	Linalool	0.21	-	0.40	-	0.11	0.31	0.12	0.19	-	0.20
13.78	α-Campholena	-	-	0.11	-	-	-	-	-	-	-
16.14	(−)-Camphor	-	-	10.43	-	-	5.36	-	5.10	-	5.32
17.01	Isoborneol	-	-	1.52	-	-	0.77	-	0.71	-	0.75
17.54	Borneol	-	-	2.71	-	-	1.38	-	1.32	-	1.39
18.00	(−)-Terpinen-4-ol	0.28	-	0.45	43.94	0.17	0.38	20.85	0.26	20.94	22.02
18.98	α-Terpineol	-	0.16	2.31	3.61	0.11	1.24	1.75	1.22	1.81	3.03
25.18	α-Bisabolol	-	-	-	0.10	-	-	-	-	-	-
26.98	4,4-Dimethylpent-2-enal	-	-	-	0.11	-	-	-	-	-	-
27.24	δ-Elemene	1.00	-	-	-	0.44	0.45	0.51	-	-	-
29.77	α-Copaene	2.22	-	-	0.10	1.01	1.08	1.24	-	-	-
30.57	β-Cubebene	0.18	-	-	-	-	-	-	-	-	-
31.67	α-Gurjunene	-	-	-	0.31	-	-	-	-	-	-
32.48	Caryophyllene	24.56	-	4.92	0.39	11.50	14.62	13.10	2.36	-	2.73
33.64	(+)-Aromadendrene	-	-	-	1.00	-	-	0.44	-	0.41	0.44
34.69	α-Caryophyllene	1.35	-	-	-	0.60	0.69	0.73	-	-	-
34.96	Aromadendrene	-	-	-	0.38	-	-	0.12	-	0.15	0.15
35.75	Isoledene	-	-	-	0.21	-	-	-	-	-	-
36.76	Pimarinal	-	-	-	-	-	-	0.11	-	-	-
36.89	Viridiflorene	-	-	-	0.80	-	-	0.36	-	0.34	0.33
37.18	γ-Elemene	-	-	-	0.22	-	-	-	-	-	-
37.43	α-Amorphene	0.24	-	-	-	-	-	-	-	-	-
37.98	β-Bisabolene	0.44	-	-	-	0.17	-	-	-	-	-
38.59	δ-Cadinene	0.60	-	-	0.87	0.24	0.26	0.75	-	0.38	0.39
42.33	Caryophyllene oxide	2.62	-	-	-	1.17	1.24	1.37	-	-	-
42.58	Globulol	-	-	-	0.33	-	-	0.14	-	0.11	0.11
43.06	Ledol	-	-	-	0.14	-	-	-	-	-	-
45.01	α-Cedrene	-	-	-	0.16	-	-	-	-	-	-
45.12	Spathulenol	0.17	-	-	-	-	-	-	-	-	-

^a^ Relative retention time (min); ^b^ Letters refer to essential oils, for the essential oils: black pepper (B), eucalyptus (E), rosemary (R), and tea tree (T). For the essential oil combinations: black pepper + eucalyptus (BE), black pepper + rosemary (BR), black pepper + tea tree (BT), eucalyptus + rosemary (ER), eucalyptus + tea tree (ET) and tea tree + rosemary (TR). (-) referred to none detected.

**Table 2 molecules-26-03055-t002:** Mortality of aphids *Myzus persicae* after contact treatment with four pure essential oils.

EO ^a^	Concentration (μL/mL)	Mortality ± SD ^b^ (%) at Different Time (h) of Treatment				
		1	3	6	8	24
B	0	0.0 ± 0.0	0.0 ± 0.0	0.0 ± 0.0	0.0 ± 0.0	0.0 ± 0.0
	1	0.0 ± 0.0	1.6 ± 2.8	5.0 ± 5.0	16.6 ± 4.0	25.0 ± 5.0
	2	0.0 ± 0.0	1.6 ± 2.8	6.6 ± 2.8	16.6 ± 5.7	26.6 ± 2.8
	3	1.6 ± 2.8	10.0 ± 5.0	21.6 ± 2.8	41.6 ± 2.8	66.6 ± 5.7
	5	8.3 ± 2.8	8.3 ± 2.8	20.0 ± 5.0	53.3 ± 7.2	80.0 ± 5.0
E	0	0.0 ± 0.0	0.0 ± 0.0	0.0 ± 0.0	0.0 ± 0.0	0.0 ± 0.0
	1	0.0 ± 0.0	1.6 ± 2.8	5.0 ± 2.9	13.3 ± 2.8	16.6 ± 2.8
	2	0.0 ± 0.0	0.0 ± 0.0	6.6 ± 2.8	13.3 ± 2.8	18.3 ± 2.8
	3	5.0 ± 5.0	10.0 ± 5.0	16.6 ± 2.8	33.3 ± 2.8	55.0 ± 0.0
	5	5.0 ± 0.0	8.3 ± 2.8	16.6 ± 7.6	33.3 ± 2.8	53.3 ± 2.8
R	0	0.0 ± 0.0	0.0 ± 0.0	0.0 ± 0.0	0.0 ± 0.0	0.0 ± 0.0
	1	0.0 ± 0.0	0.0 ± 0.0	5.0 ± 2.9	11.6 ± 2.8	23.3 ± 2.8
	2	0.0 ± 0.0	1.6 ± 2.8	6.6 ± 2.8	15.0 ± 5.0	23.3 ± 5.7
	3	3.3 ± 2.8	8.3 ± 2.8	15.0 ± 5.0	31.6 ± 2.8	56.6 ± 2.8
	5	8.3 ± 2.8	8.3 ± 2.8	23.3 ± 2.8	35.0 ± 5.0	60.0 ± 5.0
T	0	0.0 ± 0.0	0.0 ± 0.0	0.0 ± 0.0	0.0 ± 0.0	0.0 ± 0.0
	1	0.0 ± 0.0	0.0 ± 0.0	5.0 ± 0.0	6.6 ± 2.8	13.3 ± 2.8
	2	0.0 ± 0.0	0.0 ± 0.0	5.0 ± 0.0	8.3 ± 2.8	16.6 ± 7.6
	3	5.0 ± 5.0	8.3 ± 2.8	20.0 ± 5.0	45.0 ± 5.0	65.0 ± 5.0
	5	8.3 ± 2.6	13.3 ± 2.8	16.6 ± 2.8	45.0 ± 5.0	80.0 ± 5.0

^a^ Letters refer to essential oils (EOs): black pepper (B), eucalyptus (E), rosemary (R), and tea tree (T). ^b^ Standard deviation.

**Table 3 molecules-26-03055-t003:** Mortality of aphids *M. persicae* after contact treatment with mixtures of six essential oils.

EO ^a^	Concentration (μL/mL)	Mortality ± SD ^b^ (%) at Different Time (h) of Treatment			
		1	3	8	24
BE	0	0.0 ± 0.0	0.0 ± 0.0	0.0 ± 0.0	0.0 ± 0.0
	1	1.6 ± 2.8	1.0 ± 5.0	26.6 ± 7.2	51.6 ± 7.6
	2	5.0 ± 5.0	13.3 ± 5.0	3.0 ± 5.7	53.3 ± 8.6
	3	1.0 ± 0.0	13.3 ± 2.8	38.3 ± 2.8	61.6 ± 1.6
	5	11.6 ± 1.6	21.6 ± 1.7	4.0 ± 0.0	65.0 ± 2.8
BR	0	0.0 ± 0.0	0.0 ± 0.0	0.0 ± 0.0	0.0 ± 0.0
	1	1.6 ± 2.8	1.0 ± 5.0	16.6 ± 2.8	5.0 ± 5.0
	2	5.0 ± 2.8	15.0 ± 2.8	45.0 ± 2.8	55.0 ± 2.8
	3	6.6 ± 2.8	2.0 ± 2.8	45.0 ± 6.6	7.0 ± 5.0
	5	11.6 ± 1.6	2.0 ± 3.3	45.0 ± 4.4	71.6 ± 5.3
BT	0	0.0 ± 0.0	0.0 ± 0.0	0.0 ± 0.0	0.0 ± 0.0
	1	13.3 ± 2.8	25.0 ± 5.7	46.6 ± 2.8	76.6 ± 2.8
	2	11.6 ± 0.0	26.6 ± 0.0	43.3 ± 0.0	83.3 ± 0.0
	3	16.6 ± 2.8	3.0 ± 5.0	5.0 ± 2.8	98.3 ± 1.6
	5	16.6 ± 1.6	33.3 ± 4.4	65.0 ± 5.0	98.3 ± 1.6
ER	0	0.0 ± 0.0	0.0 ± 0.0	0.0 ± 0.0	0.0 ± 0.0
	1	0.0 ± 0.0	3.3 ± 2.8	11.6 ± 2.8	4.0 ± 5.0
	2	3.3 ± 2.8	16.6 ± 5.7	46.6 ± 2.8	55.0 ± 5.7
	3	5.0 ± 0.0	16.6 ± 2.8	51.6 ± 1.6	78.3 ± 4.4
	5	6.6 ± 1.9	18.3 ± 1.6	53.3 ± 4.4	76.6 ± 1.6
ET	0	0.0 ± 0.0	0.0 ± 0.0	0.0 ± 0.0	0.0 ± 0.0
	1	8.3 ± 2.8	31.6 ± 7.2	41.6 ± 11.5	7.0 ± 7.6
	2	11.6 ± 2.8	25.0 ± 0.0	7.0 ± 8.6	75.0 ± 5.0
	3	15.0 ± 2.8	26.6 ± 2.8	78.3 ± 2.8	93.3 ± 5.0
	5	18.3 ± 1.6	31.6 ± 2.8	78.3 ± 3.4	95.0 ± 5.0
TR	0	0.0 ± 0.0	0.0 ± 0.0	0.0 ± 0.0	0.0 ± 0.0
	1	1.6 ± 2.8	8.3 ± 5.7	13.3 ± 10.4	35.0 ± 8.6
	2	15.0 ± 5.0	28.3 ± 2.8	71.6 ± 2.8	86.6 ± 2.8
	3	15.0 ± 5.7	33.3 ± 4.4	8.0 ± 3.3	96.6 ± 1.6
	5	18.3 ± 1.6	35.0 ± 2.8	81.6 ± 2.8	98.3 ± 1.6

^a^ For the essential oil (EO) combinations: black pepper + eucalyptus (BE), black pepper + rosemary (BR), black pepper + tea tree (BT), eucalyptus + rosemary (ER), eucalyptus + tea tree (ET) and tea tree + rosemary (TR). ^b^ Standard deviation.

**Table 4 molecules-26-03055-t004:** Synergistic interaction of six binary combinations of four essential oils against aphids *M. persicae* after 24 h contact application.

EO	LC_50_ (95% CI) ^a^	Intercept	Slope	df ^b^	X^2 c^	H ^d^	Expected LC_50_ ^e^			
							H&P ^f^	Wadley ^g^	R ^h^	S ^i^
B	5.16 (4.57–5.75)	2.00	−0.39	58	1.41	0.02				
E	8.27 (6.39–10.14)	1.94	−0.23	58	4.42	0.07				
T	5.03 (4.56–5.50)	2.39	−0.47	58	7.64	0.13				
R	7.76 (6.05–9.47)	1.81	−0.23	58	2.12	0.03				
BE	11.86 (3.20–20.53)	1.02	−0.09	58	0.17	0.00	5.43	6.35	0.54	Add
BT	1.57 (1.28–1.88)	0.79	−0.50	58	0.65	0.01	3.81	5.09	3.24	Syn
BR	7.45 (4.85–10.05)	1.09	−0.15	58	3.95	0.06	5.17	6.2	0.83	Add
ET	2.36 (2.00–2.71)	0.86	−0.36	58	1.97	0.03	4.58	6.26	2.65	Syn
ER	5.4 (4.41–6.39)	1.30	−0.24	58	10.41	0.17	5.95	8.01	1.48	Add
TR	2.23 (2.04–2.42)	1.86	−0.83	58	29.39	0.50	5.14	6.1	2.74	Syn

^a^ Lethal concentration (LC_50_) ± Confidence Interval (CI); ^b^ df = Degree of freedom; ^c^ X^2^ = Chi square; ^d^ H = Heterogeneity factors; ^e^ Expected LC_50_ based on each calculation model; ^f^ H&P = Hewlett and Plackett’s calculation of expected LC_50_; ^g^ Wadley’s calculation of expected LC_50_; ^h^ *R* = Determination of interaction of the mixture based on Wadley’s determination method: when *R* > 1.5, synergistic (Syn) interaction; when 1.5 ≥ *R* > 0.5, additive (Add) interaction; when *R* ≤ 0.5, antagonistic (Anta) interaction; ^i^ S = Synergy ratio from Wadley’s calculation.

**Table 5 molecules-26-03055-t005:** List of essential oils used in this study, including origin and plant part used.

Essential Oil Name	Plant Family	Origin	Plant Tissues Used
Black pepper	*Piperaceae*	India	Berries
Eucalyptus Blue Gum	*Myrtaceae*	Australia	Wood and Leaves
Rosemary	*Lamiaceae*	Spain	Herb
Tea tree	*Myrtaceae*	Australia	Leaves

## Data Availability

Data are available from the authors on request.

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
