# Peer review of "Evaluation of Aphicidal Effect of Essential Oils and Their Synergistic Effect against Myzus persicae (Sulzer) (Hemiptera: Aphididae)"

_molecules, 2021, doi:10.3390/molecules26103055_

Round 1
Reviewer 1 Report
Introduction
Results
2.2. Lines – 121 - 8.33 (1 and 2 h)
2.3. Lines 127-129 - Did all the oils tested cause more than 80% mortality?
Statistical analyses include standard deviation (SD) (Tab. 2,3) and standard error (SE) (tab. 4)- one option should have been used.
How can the authors explain these high values of standard deviation, mainly at low concentrations?
2.4. line 150- BE and BR
Table 4. - Please add to title of table „after 24 h”
2.5. Line 163- The authors examined the stability of essential oils and their mixtures over a period of 1 to 12 months. Figures 1-6 present 1, 2 and 3 months (I suppose). The description of the results is for BE, but what for the others.
lines – 181 -183 – This sentence should be moved to the Introduction.
The Figures 1-6 are not clear, so I suggest including more information in their titles. I propose: „FTIR analysis for the combination of essential oil black pepper + eucalyptus (BE) in different temperature (15,25,35OC) and different time of storage (1-3 months).
Material and methods
4.1.
Table.5. " Method of extraction” - remove the column. This information is in the text and on all tested oils.
4.4. What substance was a solvent ?
Line 375 Reference?
4.7 - I did not find the cited item in the References [43].
References
The order of references section has to be corect (from 12 items).
Author Response
Response to Reviewer 1 Comments
Point 1: 2.2. Lines – 121 - 8.33 (1 and 2 h)
Response 1: Thank you, corrected.
Point 2: 2.3. Lines 127-129 - Did all the oils tested cause more than 80% mortality?
Response 2: Thank you, edited and corrected.
Point 3: Statistical analyses include standard deviation (SD) (Tab. 2,3) and standard error (SE) (tab. 4)- one option should have been used.
How can the authors explain these high values of standard deviation, mainly at low concentrations?
Response 3: Thank you, the high value of SD at low concentration maybe due to the empirical rule and also depend on the sample size (20-30 aphids per replicate).
Standard error was changed to standard deviation in Table 4 as request.
Point 3: 2.4. line 150- BE and BR
Table 4. - Please add to title of table, after 24 h”
Response 3: Thank you, corrected and the title of table 4 added “after 24 h”.
Point 4: 2.5. Line 163- The authors examined the stability of essential oils and their mixtures over a period of 1 to 12 months. Figures 1-6 present 1, 2 and 3 months (I suppose). The description of the results is for BE, but what for the others.
Response 4: Thank you, the description of the FTIR results were edited accordingly
Point 5: lines – 181 -183 – This sentence should be moved to the Introduction.
Figures 1-6 are not clear, so I suggest including more information in their titles. I propose FTIR analysis for the combination of essential oil black pepper + eucalyptus (BE) in different temperature (15,25,35OC) and different time of storage (1-3 months).
Response 5: Thank you, line 181 – 183 we cannot move to the Introduction section because this sentence from FTIR results.
For the title of 1-6 figures edited as your suggestion.
Point 6: 4.1.
Table.5. " Method of extraction” - remove the column. This information is in the text and on all tested oils.
4.4. What substance was a solvent?
Response 6: Thank you, the method of extraction column was removed from Table 5.
The solvent was Hexane.
Point 7: Line 375 Reference?
Response 7: Thank you, reference added.
Point 7: 4.7- I did not find the cited item in the References [43].
Response 7: Thank you, edited the reference.
Point 8: The order of the references section has to be correct (from 12 items).
Response 8: Thank you, the references corrected.
Reviewer 2 Report
I was pleased to learn more about the interesting and important aspect of essential oils applied to combat green peach aphid.
The evaluation of the use of essential oils to combat insects that infest plants is an important objective to be pursued in order to reduce the use of synthetic insecticides with a view to sustainable agriculture.
The authors did a good job and produced good results; however I believe that with some precautions the manuscript can guarantee a better expression of the results obtained.
It can be improved:
Row 11. These authors have the same contribution. Please check the journal guidelines if this statement is correct, also because at the end there are the Author's Contributions (row 432-435).
introduzione
Well done and well structured introduction that provides a complete picture of the work and the reader. Just a hint, supplement with a brief hint regarding FTIR technology.
Result
Row 90. Since the whole paragraph refers to the results shown in table 1, I suggest to insert (table 1) in line
86 immediately after the words "GC-MS analysis" for a better understanding.
Row 116. Move (table 2), at the beginning of the paragraph, for a better understanding of the results.
Row 122-124. “For the use of these oils, it can be concluded that black pepper and tea tree have stronger insecticidal potential than eucalyptus and rosemary”; I think this part is more appropriate in the discussion chapter.
Row 126-127. “In the series of concentrations applied, mortality was concentration dependent”; I think this part is more appropriate in the discussion chapter.
Row 162-166. “The essential oil mixtures were stored at three different temperature (15, 25 and 35 ËšC) for one, two, three, six and twelve months, and all mixtures introduce to FTIR analysis to determine functional groups changes during the storage period. The functional groups present in the mixture of essential oils were determined by comparing the vibration fre-quencies in the wave number”; I think this part is more appropriate in the Materials and Methods chapter.
Row 179-183. “All these peaks of function group were not affected by storage temperature tested on essential oils or the time of storage, and all groups kept at the same wavenumber location. FTIR analysis was performed because is a fast and relatively cheap technique that might allow direct function groups measurement of components in mixtures” I think this part is more appropriate in the discussion chapter.
Discussion
Row 197. “…use of single essential oil [26]”; indicate relevant bibliography
Row 200-201. “Caryophyllene 24.56% was the most abundant major constituents 200 followed by D-Limonene 15.52%, α-Pinene 12.66%, β-Pinene 12.17%, and Sabinene 8.6%”; considering that the percentages have been reported in the results chapter, the compounds could be indicated without reporting the percentages or the percentages can be reported inside the brackets.
Row 207. “Eucalyptol 35.27%, α-Pinene 15.87%, (-)-Camphor 10.43% , and β-Pinene 8.5%”; considering that the percentages have been reported in the results chapter, the compounds could be indicated without reporting the percentages or the percentages can be reported inside the brackets.
Row 207-209. “These components have pesticide action that is used in pest control as presented in many past experiments [14,26]”; I think it is appropriate to integrate with other bibliography.
Row 210. “…(–)-Terpinen-4-ol 43.94% and γ-Terpinene 17.74%”; considering that the percentages have been reported in the results chapter, the compounds could be indicated without reporting the percentages or the percentages can be reported inside the brackets.
Row 214-215. “…Caryophyllene 11.50%, D-Limonene 10.58%, α-Pinene 6.76%, β-Pinene 214 6.12%, and Sabinene 4.22%”; considering that the percentages have been reported in the results chapter, the compounds could be indicated without reporting the percentages or the percentages can be reported inside the brackets.
Row 221-222. “…α-Pinene 14.60%, β-Pinene 10.43%, 221 D-Limonene 9.34%, Eucalyptol 19.63%, and Caryophyllene 14.62%”; considering that the percentages have been reported in the results chapter, the compounds could be indicated without reporting the percentages or the percentages can be reported inside the brackets.
Row 241-242. “The essential oil constituents vary from type to other depending on plant species.” The bibliography should be included.
Row 242. “[30,31] reported that a-Pinene, D- 242 Limonene and Camphene, which are….”; in my opinion it would be more appropriate to insert Hollingsworth et al., [30] and Liška et al., [31] reported that a-Pinene, D-Limonene and Camphene, which are major constituents…..
Row 252-254. “Rosemary oil has insecticidal activities against several insects and is used in many commercial products as insecticides”; The bibliography should be included.
Row 254. “A study conducted by [35] found that rosemary ….“ it would be appropriate to insert the name: A study conducted by Digilio et al., [35] found that rosemary….
Row 256-257. “These findings are similar with the results presented by [36], indicating an increase….”; it would be appropriate to insert the name: These findings are similar with the results presented by Tomova et al., [36], indicating an increase…
Row 264. “94.44 and 95.56%, respectively (Tables 2 and 3)”; Assigns the reference tables to the various data.
Row 288. “This findings are parallel to [41] who reported…”; indicate the author for a better understanding.
Materials and Methods
Row 375. "Equation error! Reference source not found. And error! Reference source not found." What does this mean ?, is it a typo?.
Row 376. "[26]"; indicate the relevant bibliography.
Conclusion
The conclusions are good and well articulated, they reflect the hypothesis described in the introduction.
Reviewer 3 Report
General comments
This manuscript reports results on the toxicity of four essential oils and their binary combinations to the important pest Myzus persicae under laboratory conditions, as well as the characterization of the composition of the essential oils and the combinations tested. The objectives of the manuscript are clearly stated, but the authors say: “In addition, it evaluates a synergistic interaction between binary mixtures of essential oils against M. persicae in different methods of application” and, however, only one test type in described in the methodology. It may be a question of wording, as the test type carried out includes contact and residual effects. In fact, authors refer sometimes to “contact and residual bioassays methods” (section 2.2., lines 111-112) and sometimes to “contact toxicity” (as in the heading of the same section 2.2.). This must be clarified. The experimental design is appropriate and the data analysis using the probit analysis is the commonest approach used in these kind of experiments. I have some comments on this question in the Specific Comments section. The statistical analyses section says “Mortality data from the essential oil elimination assay….”, but any essential oil elimination assay is described in the manuscript and no results of any ANOVA are reported. The results are clearly shown, but, as an agricultural entomologist, I am not qualified to judge on the FTIR analysis and results. The results do not show that any essential oil or any combination is significantly more active against M. persicae than any other one, as the intervals of the LC50 values of the 10 probit lines calculated overlap, which means that they are not significantly different. The Discussion section in not properly ordered, as some ideas are presented repeatedly, and could be clearly shortened and improved. To my opinion, authors overestimate the degree of the activity of the essential oils in their conclusions. I would not qualify as “strong” the insecticidal effect of any essential oil alone for two reasons; first, the maximum mean mortality observed in the experiments 24 h after the treatment was 80 %; second, a positive control (an insecticide with known strong effect on M. persicae) is necessary to make this affirmation. Only the binary combinations having tea tree as one of its components produced a mortality higher than 95 %.
In conclusion, the reported results could be of interest due to the importance of M. persicae as a pest and its ability to develop resistance to insecticides, but the observed efficacy of the essential oils testes is medium in most of the cases and any essential oil as shown an efficacy higher than the efficacy of any other one. This is the main reason to recommend rejecting this manuscript for this journal.
Specific comments
Lines 33-34. Mention also fruit crops, peach trees, for example, as M. persicae hosts.
Comments on tables 2 and 3:
- Standardize the number of decimal figures. One decimal figure is enough. Include the SD (correct Division for Deviation in the footnote of the tables) when all the values of the replicates are equal (i.e. 0.0 ±0), as zero is not a missing value.
- The heading of the columns 1 – 3 – 6 – 8 – 24 should be “Time from insecticide application (h)”, instead of “Mortality …”, what is said in the table heading.
- Table 3 does not show the results of “six essential oils”, but of “six binary combinations of four essential oils”.
Comments on table 4
- The heading is not correct, as Table 4 does not show the mortality, but the results of the probit analysis.
- Change LD by LC, as concentrations, and not doses, are applied in the experiments.
Comments on the probit analyses and their results
- The heterogeneity factor values of each regression line (Chi-square value divided by the number of degrees of freedom) should be presented and their values compared to 1.
- The intervals of the LC50 values of the 10 probit lines calculated overlap, which means that they are not significantly different.
Line 362. Mention the aphis stage (wingless adult females?) or instar (young, mature nymphs?) used in the experiments
Round 2
Reviewer 3 Report
I very much appreciate the improvements the authors have included in the manuscript and I think that the manuscript could be published in Molecules if the editor decides it. I am sorry, but I have to insist on some problems I still find in the manuscript.
My main concern relates to the presentation of the probit analyses results and their interpretation:
- I suggested the authors to include the heterogeneity factor values of each regression line (Chi-square value divided by the number of degrees of freedom). The authors have not accepted this suggestion, what is their right, of course, without giving any reason for it. When the heterogeneity factor is greater than 1.0, a plot of the data should be examined because the data do not fit the model. Such a plot may reveal systematic departure from linear regression, in which case a function other than logarithm of dose may be more appropriate.
- I wrote that “the intervals of the LC50 values of the 10 probit lines calculated overlap, which means that they are not significantly different”, which means that one cannot conclude that one EO or one EO binary combination is significantly more effective than others. The authors do not comment on that point.
- The authors have substituted the Standard Error for the Standard Deviation in Table 4. The values of the SD presented in the modified Table 4 are smaller than the values of the SE presented in the original Table 4, what is not possible, as the value of the SD is always greater than the value of the SE.
- I have detected one error in Table 4 that I did not detected previously, I apologize. For BE and BR the value of the LC50 is outside the Confidence Interval, what is not possible.
The second concern related to the conclusions:
- The mortality caused by eucalyptus and rosemary EO at the highest concentration tested and at 24 h after their application is 53 % and 60 %, respectively. In my opinion, it cannot be qualified as a “strong toxic effect”.
- Contact and residual toxicity is still mentioned in the conclusions section (lines 1083-1084).
- Tea tree EO was more effective than eucalyptus EO (line 1085).
Only minor changes have been done in the Discussion section. The authors have not answered to the comment on the original Discussion section.
Author Response
manuscript ID molecules-1164655
Entitled “Evaluation of Aphicidal Effect of Essential Oils and their Synergistic effect against Myzus persicae (Sulzer) (Hemiptera: Aphididae)”
Response to Reviewer 3 Comments
Thank you for your response and fruitful comments. We found the comments helpful. The data has been reanalyzed by statistician Dr Penghao Wang and added as a co-author. Hopefully, we have incorporated the suggested improvements to your satisfaction. The list of addressed review comments point-by-point to the reviewer 3 comments is attached as below.
Point 1: I suggested the authors to include the heterogeneity factor values of each regression line (Chi-square value divided by the number of degrees of freedom). The authors have not accepted this suggestion, what is their right, of course, without giving any reason for it. When the heterogeneity factor is greater than 1.0, a plot of the data should be examined because the data do not fit the model. Such a plot may reveal systematic departure from linear regression, in which case a function other than logarithm of dose may be more appropriate.
Response 1: Thank you, we accordingly revised to emphasize this point. We have re-performed the calculation focusing on after 24 hour treatment. Hewlett and Plackett’s model and Wadley’s model were used to compare expected and observed LC50 values to show the synergistic interaction between essential oils as described in the manuscript. We added the Chi square, degree of freedom and the heterogeneity values for each oil to the Table 4 as recommended.
Point 2: I wrote that “the intervals of the LC50 values of the 10 probit lines calculated overlap, which means that they are not significantly different”, which means that one cannot conclude that one EO or one EO binary combination is significantly more effective than others. The authors do not comment on that point.
I have detected one error in Table 4 that I did not detected previously, I apologize. For BE and BR the value of the LC50 is outside the Confidence Interval, what is not possible.
Response 2: Thank you, we presented our results for the use of essential oils (EO) and reported the mortality according to the concentration of EO that used. Our results showed the interaction of EO depended on statistic models that used as shown in Table 4; a synergistic interaction appeared in the one combination of essential oils as shown in the results. Based on Hewlett and Plackett’s model and Wadley’s model to compare expected and observed LC50 values. We reviewed and edited this point in the manuscript with the word track.
For the Confidence Interval for BE and BR, we have re-performed the analysis and results shown in the manuscript Table 4.
Point 3: The authors have substituted the Standard Error for the Standard Deviation in Table 4. The values of the SD presented in the modified Table 4 are smaller than the values of the SE presented in the original Table 4, what is not possible, as the value of the SD is always greater than the value of the SE.
Response 3: Thank you; we have re-done the analysis and re-calculated everything to make sure the results are correct. Please see revised Table 4 for this point.
Point 3: The second concern related to the conclusions:
The mortality caused by eucalyptus and rosemary EO at the highest concentration tested and at 24 h after their application is 53 % and 60 %, respectively. In my opinion, it cannot be qualified as a “strong toxic effect”.
Response 3: Thank you, edited and done.
Point 4: Contact and residual toxicity is still mentioned in the conclusions section (lines 1083-1084).
Response 4: Thank you, done.
Point 5: Tea tree EO was more effective than eucalyptus EO (line 1085).
Response 5: Thank you, done.
Point 6: Only minor changes have been done in the Discussion section. The authors have not answered to the comment on the original Discussion section.
Response 7: Thank you for your suggestions and your feedback. However, we clarified every previously point in your feedback.
In addition to the above comments, all comments and feedback pointed out by the reviewers have been corrected in the manuscript.